# A field based study of swimbladder adjustment in a physostomous teleost fish

Kostas Ganias[1], Stella Michou[1] and Cristina Nunes[2]

[1] School of Biology, Aristotle University of Thessaloniki, Thessaloniki, Greece
[2] IPMA—Instituto Português do Mar e da Atmosfera, Lisboa, Avenida de Brasilia s/n, Portugal

## ABSTRACT

The present study assesses swimbladder dynamics in natural occurring sardine, *Sardina pilchardus*, populations with the aim to examine whether this is affected by bathymetric positioning and the physiological state of the individuals. To do so, swimbladder size and shape were modeled in relation to catch depth and the size of various visceral compartments such as gonad, liver, fat and stomach. Swimbladder size was shown to be related to depth in a way that individuals with smaller swimbladders occurred at larger depths. Moreover, evidence is provided that the swimbladder in sardine might have a functional relationship both with the reproductive and the feeding state of individuals, since none of the fish with hydrated gonads and/or large stomachs displayed distended swimbladders.

## INTRODUCTION

In most marine teleosts the swimbladder acts primarily as a hydrostatic organ, while in a few species it may also function as an organ for sound production (e.g., oyster toadfish; *Fine, McKnight Jr & Blem, 1995*). Teleosts may be either physoclists, having a closed swimbladder, or physostomes, having a swimbladder which is connected to their oesophagus via a duct, allowing uptake and release of gas through the mouth (*Stewart & Hughes, 2014*). In physostomes like clupeids, the swimbladder provides a dual function, acting as a buoyancy regulating organ and as a gas reservoir for the acoustico-lateralis system (*Nero, Thompson & Jech, 2004*). Furthermore, the fish swimbladder is the main reflector of acoustic energy, being responsible for up to 90–95% of the backscattered sound intensity which is of primary importance in acoustic estimates of fish abundance (*Foote, 1980*). As a corollary, mapping the swimbladder structural morphology and understanding factors that may affect its size and shape is essential in quantifying its contribution to several biological functions such as buoyancy regulation and in improving accuracy in estimates of fish biomass from acoustic surveys.

To date, most attempts to estimate swimbladder size have been performed by estimating its volume from the amount of gas it contains by means of gas colleting and measuring

Corresponding author
Kostas Ganias, kganias@bio.auth.gr

devices (e.g. *Blaxter, Denton & Gray, 1979*; *Fine, McKnight Jr & Blem, 1995*; *Robertson et al., 2008*). However, analysis of the structural morphology of the swimbladder needs more laborious techniques and specific equipment like the one described by *Ona (1990)* and *Machias & Tsimenidis (1995)*, who used photographs of parallel body slices to reconstruct the form of both swimbladder and other visceral compartments in order to accurately estimate their volume and shape. In another study, *Robertson et al. (2008)* visualized and measured the size of zebrafish swimbladder by means of X-ray imagery.

Despite their accuracy, the aforementioned methods are mostly designed to work under laboratory conditions, which almost prohibits their use in field surveys and consequently in the study of swimbladder dynamics in fish natural environment. The objective of the present work was to study swimbladder dynamics in wild populations of sardine, *Sardina pilchardus*. Sardine (in common with other clupeoids) are physostomes and as such their swimbladder is not closed but connected both to the alimentary canal via a valved pneumatic duct and to the anus (*Blaxter, Denton & Gray, 1979*). Due to this physiological specificity, the sardine swimbladder may be subject to volume modifications, especially during its extensive diel vertical migrations (*Zwolinski et al., 2007*), mainly due to significant changes in water pressure. In order to examine whether swimbladder dynamics in the Atlantic sardine are affected by bathymetric positioning and the physiological state of individuals, we modeled the relationship of swimbladder size and shape with depth and the size of various visceral compartments.

## MATERIALS AND METHODS

Sardine samples were collected off Portugal and the gulf of Cadiz in October 2008 and April 2009 within the remit of autumn/spring acoustic surveys carried out by the Portuguese Sea and Atmosphere Institute (IPMA, formerly IPIMAR) onboard the RV "Noruega" (Table 1). All samples were collected using either a midwater or a bottom trawl with a vertical opening of ∼10 m, towed at speeds of 3.5–4 knots for an average duration of 20 min (s.d.: 1.3). For each of the 10 samples used for the present analysis, spatial coordinates, time, bottom and fishing depth were registered. In each haul, sardines were sorted out from the remaining catch and a sample was selected in a way that all length classes would be represented. Fish were subsequently processed on deck, immediately after capture.

Each individual was sexed and measured for total length, $L$ (mm), and total weight, $W$ (0.1 g), whilst maturity, fat index and stomach fullness were scored macroscopically as follows: $P_h$ was the prevalence of females with hydrated ovaries, i.e., females that were ready to spawn; $P_f$ was the prevalence of fish with well formed layers of fat surrounding the gut (fat stage >2; see also *Silva et al., 2006*); $P_s$ was the prevalence of fish with stomachs more than half full (>stage2 from the scale described by *Cunha, Garrido & Pissarra (2005)*). Viscera were then carefully removed to avoid possible damaging of the swimbladder and eviscerated body weight was also recorded.

Subsequently, the coelomic cavity was opened either manually or using forceps in order to expose the entire cross-section of the swimbladder (Fig. 1). Swimbladders were scored macroscopically with respect to size (1: small; 2: medium; 3: large; 4: distended;
**Table 1 Sampling table.** Characteristics of the ten samples used to measure biometric data and swimbladder size and shape of sardine, *Sardina pilchardus*. $n$, number of fish analyzed; $L$, average length (cm); $SB_{xsa}$, average swimbladder ventral cross-section (cm$^2$). Minimum and maximum values are provided in parentheses.

| Sample | Date | Time | Average haul depth (m) | $n$ | $L$ | $SB_{xsa}$ |
|---|---|---|---|---|---|---|
| 1 | 18/10/2008 | 9:20 | 25,0 | 29 | 20,1 (18,2–22,2) | 4,14 (0,70–9,46) |
| 2 | 19/10/2008 | 13:45 | 17,3 | 39 | 20,3 (19–22,7) | 5,54 (1,80–12,11) |
| 3 | 20/10/2008 | 16:15 | 24,0 | 31 | 18,3 (14,7–20,6) | 2,97 (1,13–5,87) |
| 4 | 21/10/2008 | 8:45 | 45,5 | 45 | 19,1 (18,0–20,3) | 3,77 (1,92–7,50) |
| 5 | 24/10/2008 | 13:50 | 15,0 | 33 | 19,5 (18,5–20,4) | 4,05 (1,95–7,66) |
| 6 | 3/4/2009 | 17:20 | 65,0 | 11 | 19,9 (17,5–21,2) | 3,03 (2,27–4,17) |
| 7 | 1/4/2009 | 17:08 | 61,0 | 26 | 19,5 (17,6–21,4) | 2,81 (1,60–4,64) |
| 8 | 16/4/2009 | 15:57 | 26,0 | 3 | 18,9 (18,3–19,7) | 2,28 (1,19–3,36) |
| 9 | 18/4/2009 | 15:07 | 49,0 | 3 | 19,4 (19,2–19,7) | 4,71 (4,45–4,97) |
| 10 | 21/4/2009 | 18:38 | 25,0 | 2 | 21,5 (21,2–21,8) | 5,02 (3,71–6,33) |

Figs. 1A–1D) and shape (1: normal-elliptical; 2: medially compressed; 3: compressed at the anterior region; 4: compressed at the posterior region) (Figs. 1E–1G) and were then photographed using a camera supported on a vertical plastic stand. This device provided easy adjustment of camera height to allow for quick and easy framing of specimens of different sizes and was essential to avoid vibrations and to always take vertical pictures without any variation in picture angle. A ruler was always placed aside each specimen in order to calibrate scale in subsequent linear measurements. Gonads and liver from each fish were placed in plastic bags and frozen at −20 °C to be weighed (0.0001 g) in the laboratory; gonadosomatic and hepatosomatic indices were calculated using the formulas GSI = gonad weight/eviscerated weight∗100 and HSI = liver weight/eviscerated weight∗100, respectively. Biometric data and photographs from 222 sardines were collected and used for the analysis (Table 1).

The dimensional characteristics of the swimbladder were studied in more detail by processing the digital images using ImageJ (v.1.43; http://rsbweb.nih.gov/ij/). Specifically, the ventral cross-sectional area of the swimbladder ($SB_{xsa}$) was measured by spatially calibrating the image using the ruler and by drawing the perimeter of the swimbladder, $SB_p$. Swimbladder shape was also studied by measuring circularity $SB_c$: a value of 1.0

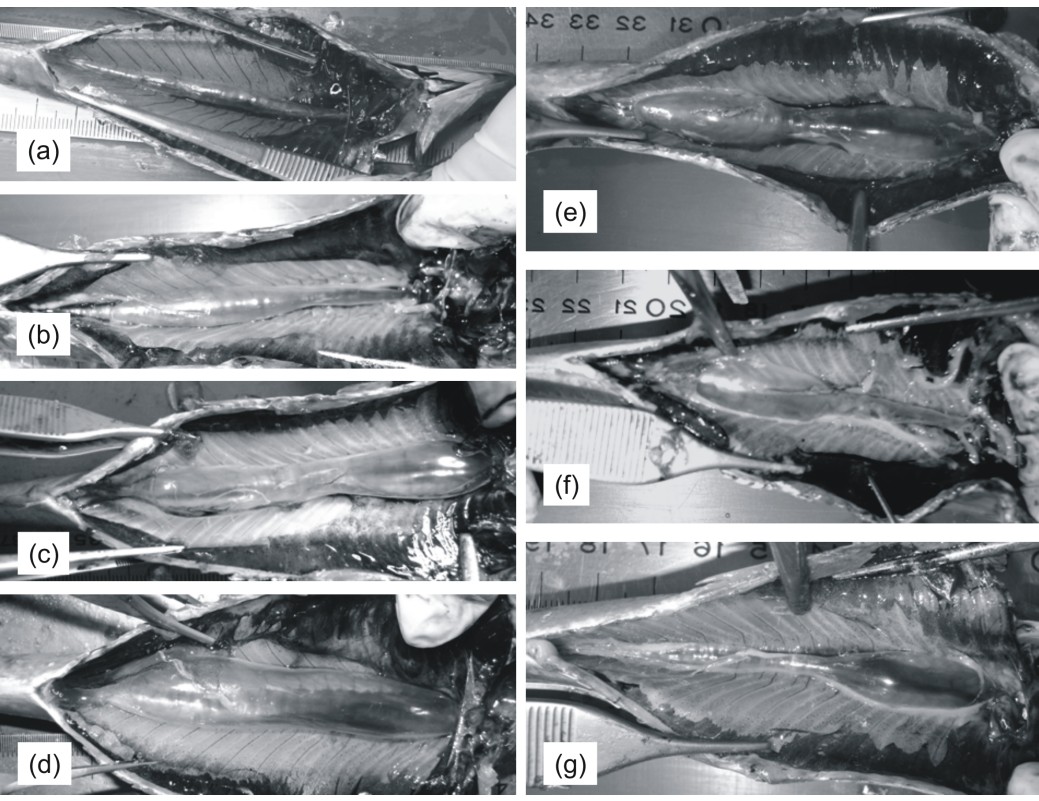

**Figure 1 Pictures of sardine swimbladders of various size and shape.** Ventral views of sardines with the abdomen opened to expose the swimbladder. (A)–(D) illustrate variability in size among swimbladders with normal-elliptical shape: (A) small; (B) medium; (C) large; (D) distended. (E)–(G) illustrates variability in shape in swimbladders with large size: (E) compressed medially; (F) compressed at the anterior region; (G) compressed at the posterior region. The head is always to the right.

indicates a perfect circle whilst lower values indicate an increasingly elongated shape. In that respect $SB_c$ together with $SB_{xsa}$ served as indices of inflation (higher values) or deflation (lower values) of the swimbladder.

The allometric relationship between $SB_{xsa}$ and $L$ was linear on a log–log scale and did not differ significantly between the two sexes (analysis of covariance, ANCOVA; slopes: $P > 0.1$; intercepts: $P > 0.1$; Fig. 2). As a consequence, a single relationship was used for the specimens and the effect of fish size from swimbladder size was removed by using the residuals, $R$, of this allometric relationship (hereinafter called as the relative swimbladder size). First, $R$ served to validate the four-scale macroscopic assignments of swimbladder size. Variability in $P_f$, $P_s$, and $P_h$, with macroscopic swimbladder size was tested using analysis of means (ANOM) while variability in $HSI$ with macroscopic swimbladder size was tested using ANOVA. Variability in $P_h$, was analysed using only the females while all other variables were analysed using both sexes. The relationship of swimbladder size ($R$) and shape ($SB_c$) with haul depth was tested by means of generalized linear models, GLMs, using in both cases, sex as a covariate. Quantile and residual inspection plots revealed that

**Peer**J

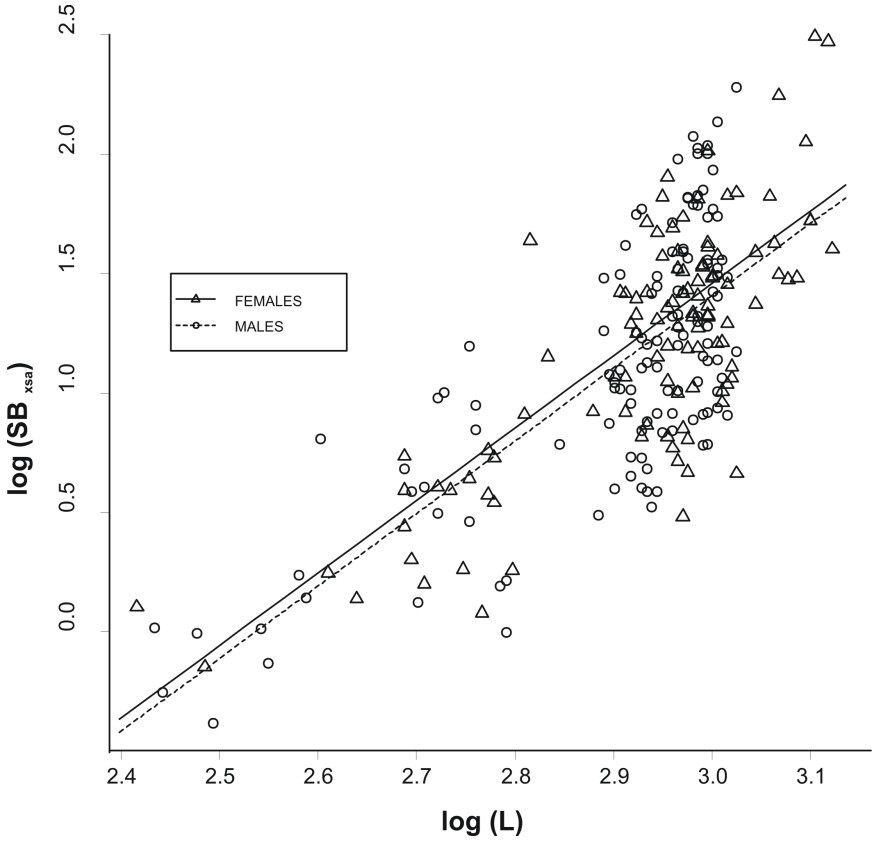

**Figure 2 Relationship between swimbladder cross-section and body length.** Relationship between the log transformed values of swimbladders' ventral cross-sectional areas ($SB_{xsa}$) and body length ($L$) for the two sexes. The linear regression lines for the two relationships are super-imposed.

a Gaussian model with an identity link was the most appropriate for the analysis of $R$ data whilst $SB_c$ data were analyzed using Gaussian models with a logarithmic link.

## RESULTS

At a first step, macroscopic assignments of swimbladder size were validated with image analysis-based measurements of cross-sectional areas; both relative swimbladder size, $R$, and circularity, $SB_c$, increased significantly in all four size levels of the macroscopic scale (Fig. 3) which validated its use in subsequent analyses. As shown in Fig. 4 most fish had medium sized swimbladders and in most cases swimbladder possessed a normal elliptical shape.

Swimbladder size was shown to be related with both the reproductive and the feeding state of the individuals. Specifically, hydrated females mainly possessed swimbladders of small or medium size while none of the females with distended swimbladder was hydrated (Fig. 5A). Similarly, the prevalence of fish with filled stomachs declined with increasing swimbladder size, and none of the individuals with distended swimbladders had its stomach more than half full (Fig. 5B). Concerning HSI and fat content there was no significant relationship with swimbladder size (HSI, ANOVA: $P > 0.1$; Fat, ANOM: $P > 0.1$).

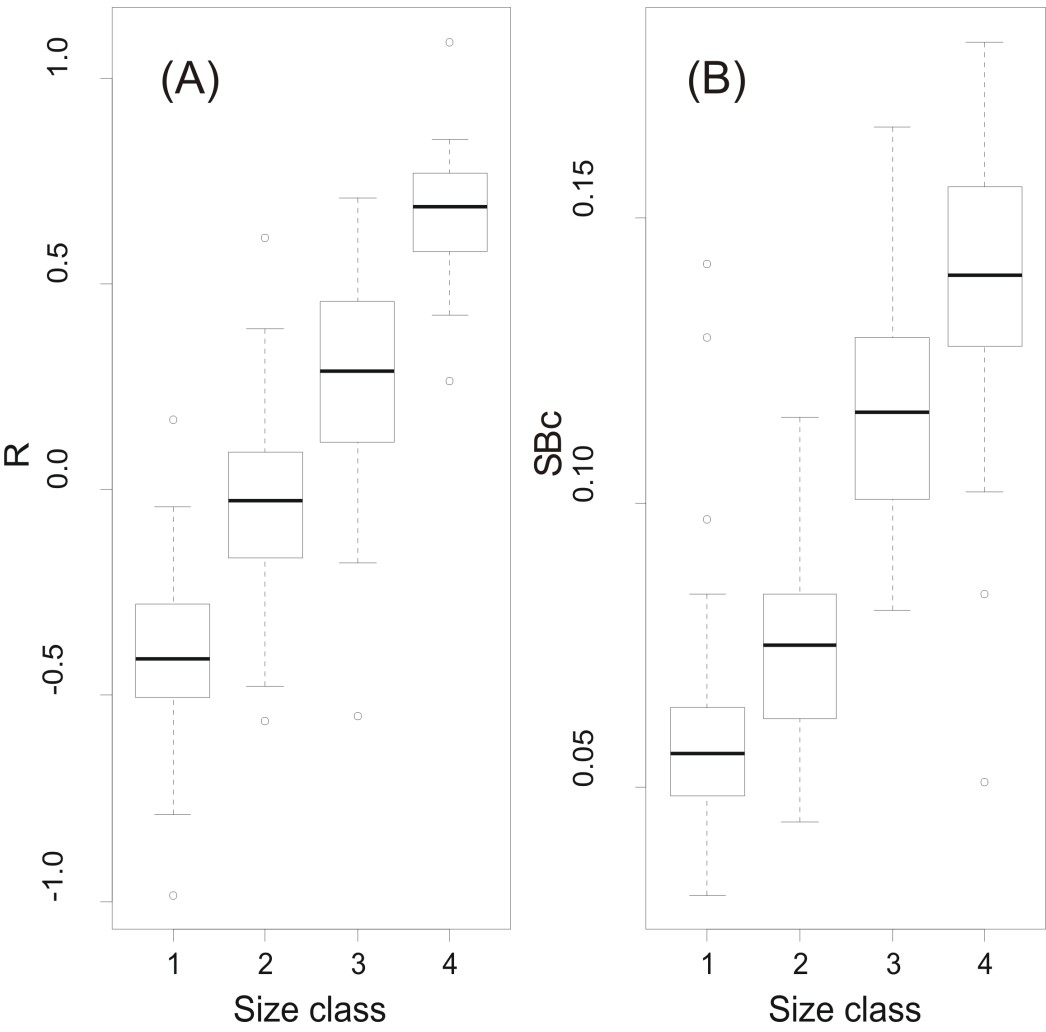

**Figure 3  Swimbladder size and shape per macroscopic size class.** Box-and-Whisker plots of (A) relative swimbladder size, $R$, and (B) circularity, SBc, with macroscopic swimbladder size classes (1: small; 2 medium; 3; large; 4: distended).

GLM analysis showed a highly significant relationship between circularity $SB_c$ and haul depth (Table 2 and Fig. 6A). The relationship between $R$ and depth was weaker whilst sex was not a significant covariate in none of the two GLMs (Table 2). Further examination of the scatter plots of swimbladder shape (Fig. 6A) and size (Fig. 6B) with haul depth suggests that small, deflated swimbladders occur at all depths while large, distended swimbladders mostly occur at smaller depths. Therefore, haul depth appears to constrain, rather than determine, the range of swimbladder sizes and fish at lower depths exhibit greater spectrum of swimbladder sizes.

## DISCUSSION

The present study provides a simple and cost-effective means for assessing swimbladder adjustment in sardine, which is a physostomous fish. *Clemens & Stevens (2003)* and

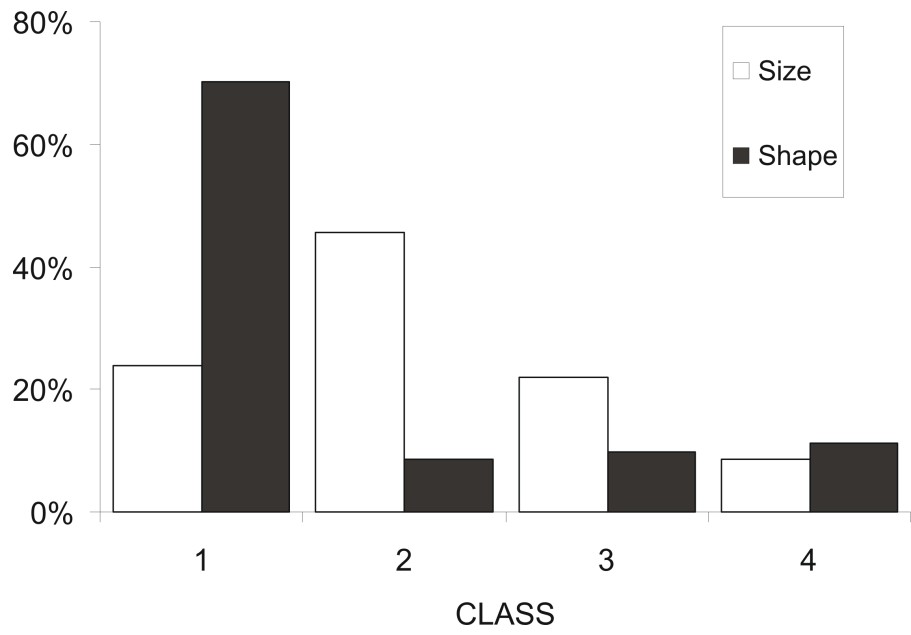

**Figure 4  Frequency distribution of sardine swimbladders per size and shape class.** Frequency distribution of sardine swimbladders at each size (white bars; 1, small; 2, medium; 3, large; 4, distended) and shape (black bars; 1, normal-elliptical; 2, medially compressed; 3, compressed at the anterior region; 4, compressed at the posterior region) class.

**Table 2  GLM model coefficients.** Coefficients of the models used to analyze the effect of depth, sex, and their interaction on sardine swimbladder size and shape. $SB_c$, swimbladder circularity; $R$, residuals of the relationship between swimbladder ventral cross-section and body length.

| Source of variation | $SB_c$ | $R$ |
|---|---|---|
| Null | 0.102** | 0.131* |
|  | (0.007) | (0.071) |
| Depth | −0.001** | −0.004* |
|  | (0.000) | (0.002) |
| Sex | ns | ns |
| Depth* Sex | ns | ns |

**Notes.**
ns, non significant.
* $0.05 > P > 0.01$.
** $P < 0.01$.

*Fässler et al. (2009)* also measured swimbladder volume in other physostomous fish, the bloater, *Coregonus hoyi*, and the Atlantic herring, *Clupea harengus*, respectively, and subsequently correlated it with the depth of capture. However, unlike our study, the authors measured swimbladder volume in the laboratory, hours after capture, through the displacement method or by magnetic resonance imaging. On the contrary, our measurements were performed on deck, immediately after capture with a semi-invasive method which did not require swimbladders to be excised.

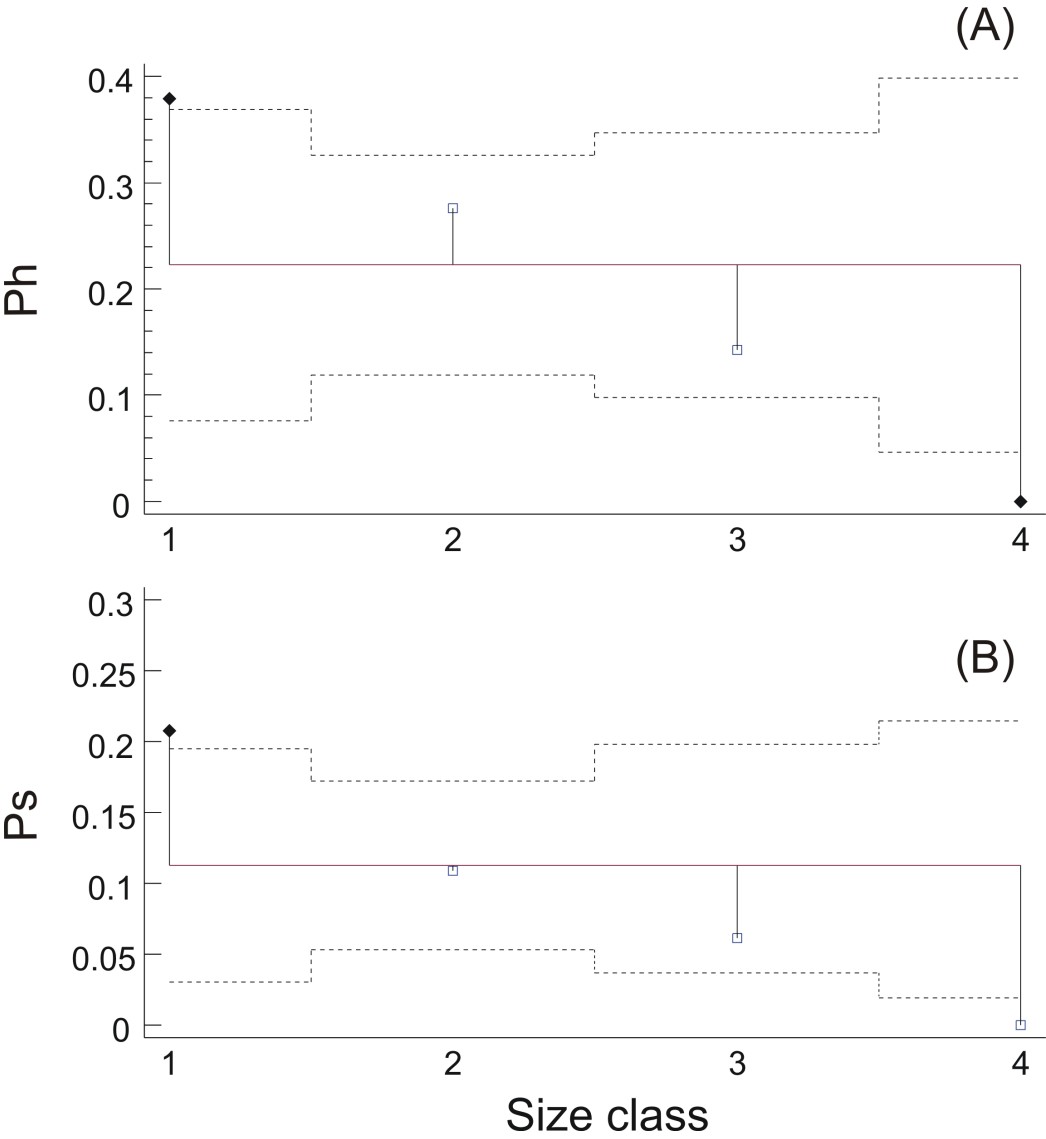

**Figure 5 Relationship between ovarian hydration and stomach fullness with swimbladder size.** (A) Prevalence of hydrated females, $P_h$, and (B) prevalence of fish with stomachs more than half full, $P_s$, in each swimbladder size class. Dotted horizontal lines represent 95% decision limits, whilst filled diamonds in the same plots represent proportions that differ significantly from the average (solid horizontal line).

There are good reasons to believe that our measurements were accurate and that our on-deck measurements were consistent with *in situ* swimbladder size values. The two main sources of bias that could have affected our measurements are (a) the effect of hauling process including the rapid ascent of fish and (b) the effect of dissection on swimbladder size and shape. *Blaxter, Denton & Gray (1979)* suggested for herring, another closely related physostomous clupeoid, that it was surprising that some of the fish could be so distended without activating the voiding reflex through the anal duct. Experiments in herring show that its swimbladder may not produce any gas, thus preventing it from actively adjusting the volume of the swimbladder (*Blaxter & Batty, 1984*). The most standing hypothesis at

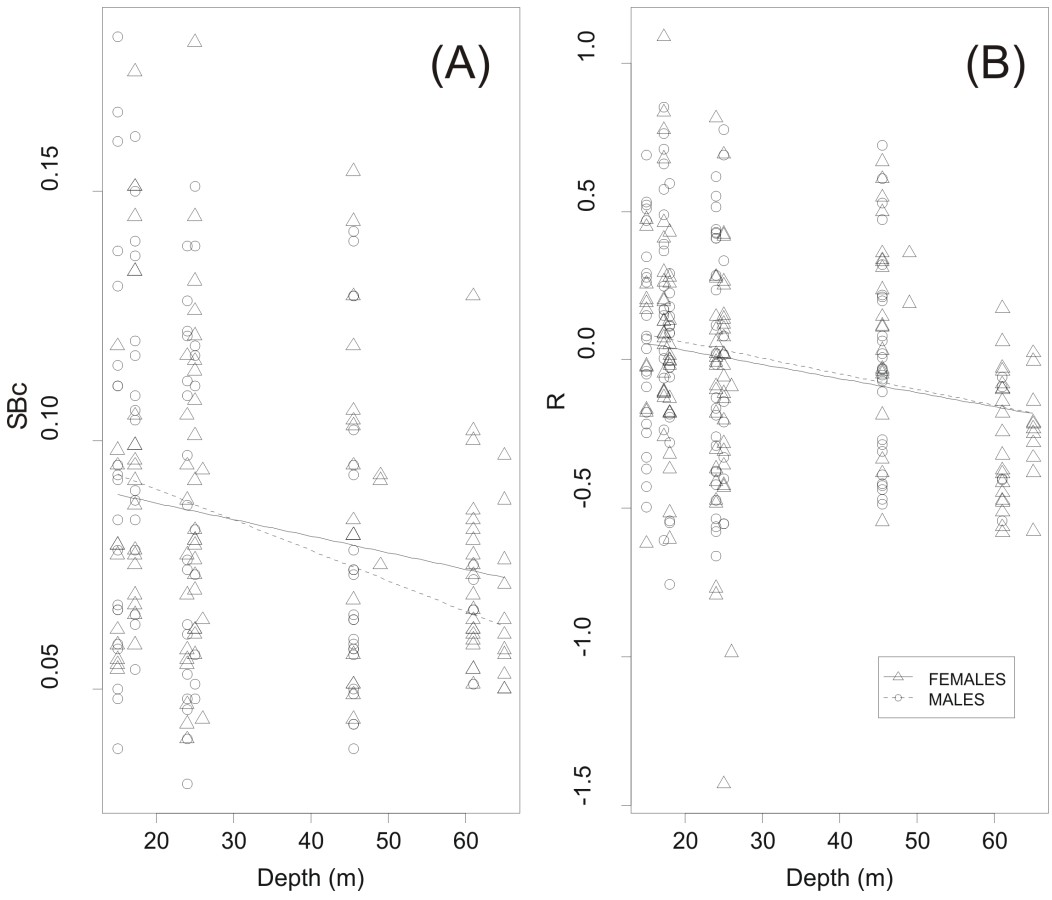

**Figure 6 Effect of haul depth on shape and size of sardine swimbladder.** Component effect of haul depth on (A) the circularity ($SB_c$) and (B) the relative size of sardine swimbladders ($R$) as derived from GLM analysis for the two sexes separately.

present is that herring is only able to inflate its swimbladder by gulping atmospheric air at the sea surface (*Ona, 1990*). Even if gas may be released during ascent, this is mostly supposed to be a behavioral rather than a physiological response, e.g., the production of bubble clouds to confuse predators (*Nottestad, 1998*). Therefore, herring swimbladder will decrease its volume with increasing ambient pressure when descending at larger depths (*Fässler et al., 2009*) but will not necessarily alter its volume when ascending. If this also happens with sardines, as suggested by *Blaxter & Hunter (1982)*, then we may hypothesize that the hauling process, which involves fish capture at a depth and subsequent ascent, will not much affect the *in situ* values of swimbladder volume. In that respect, neither the sheer physical process of trawling and hauling nor the period that fish spend in the codend, are expected to cause major alterations in swimbladder volume. In addition, fishing duration was always quite small (∼20 mins) and the ascent duration lasts no longer than 10–15 min for the depths considered in this study, so the time between catching and bringing fish to the surface would not allow sardine swimbladder to physically respond to the pressure and
temperature range of the haul. Moreover, in herring (and possibly also in sardine), the rate of gas loss by diffusion is low (*Blaxter, Denton & Gray, 1979*).

Concerning the effect of dissection, *Robertson et al. (2008)* showed for another physostomous teleost, the zebrafish *Danio rerio*, that swimbladder size in dissected specimens did not differ significantly from *in situ* X-ray measurements. Given that our anatomy scheme was semi-invasive for the swimbladder, simply involving slitting and opening of the coelomic cavity and removal of the viscera in order to expose the swimbladder, its size must have remained unaffected.

Both size and shape of the swimbladder was quite variable. Taking into account the finite volume of the coelomic cavity, the significant changes in swimbladder size from a thin strip to a distended sac and the transient changes in the size of most visceral compartments it might be postulated that swimbladder size can determine the size of some organs inside the coelomic cavity. Interestingly, swimbladder size was shown to be related only with organs that undergo transient changes (stomach: diurnal changes; gonads: daily, diurnal changes) and not with liver and fat which fluctuate at a rather seasonal scale (*Nunes et al., 2011*). Indeed, the decrease in the proportion of fish with hydrated ovaries or filled stomachs with increasing swimbladder size and more importantly the complete absence of these reproductive/feeding states in fish with distended swimbladders indicate that swimbladder size might have an adoptive or functional relationship with these two organs. *Blaxter & Batty (1984)* also observed that herrings with full stomachs and well developed gonads generally have deflated swimbladder. For instance, in fish with distended swimbladders the excess of food could be evacuated due to limitations in stomach volume inside the restricted coelomic cavity. However, *Machias & Tsimenidis (1995)* suggest for the Mediterranean sardine that the stomach has marginal or no influence on the swimbladder volume. Also, contrary to the present study, in other small-pelagic, physostomous fish (Mediterranean sardine, Barents Sea capelin, spring-spawning herring: *Machias & Tsimenidis, 1995*; *Jørgensen, 2003*; *Fässler et al., 2008*, respectively), swimbladder volume is associated to body fat content, as fat gives the fish additional buoyancy and less need for swimbladder volume (and vice-versa). For the present study, samples were collected in October and April, which correspond to periods of high and low fat content for sardine, respectively (*Nunes et al., 2011*).

On the other hand, the relationship between swimbladder size and reproductive state might be of more adoptive nature. Gonad size in sardine is known to undergo significant changes during the spawning cycle while a striking increase takes place at oocyte hydration, just previous to spawning (*Somarakis et al., 2004*). Even from simple observations it might be inferred that a pair of fully hydrated ovaries would hardly co-occur with a distended swimbladder inside the coelomic cavity. In that respect, it might be postulated that spawning at a depth where swimbladders are deflated provides more space for hydration to occur and for ovaries to develop. Indeed, *Ganias & Nunes (2011)* showed that actively spawning female sardines segregate out of the remaining population, at larger depths, taking with them large proportions of males to form ephemeral spawning aggregations. In that respect, changes in swimbladder size could be partially linked to changes in the vertical

positioning of actively spawning sardines. Accordingly, for the Mediterranean sardine, while the covariate explaining the highest proportion of the variation of the volume of the swimbladder during the species reproductive resting period was fish length, during the reproductive period the relative size of the gonad became gradually more important, the relationship of gonads to the swimbladder being equally as important as that of the swimbladder to fish length (*Machias & Tsimenidis, 1995*).

As for sex related differences, the results showed that the cross-sectional area and circularity of the swimbladder were not significantly different between males and females, and sex was not a significant explanatory variable in the model. For the Mediterranean sardine, the influence of sex on the swimbladder volume appeared as significant only for fish sampled at the beginning of the spawning season (October), because of the different degree of gonadal growth between males and females (ovaries more voluminous than testes), but not at other periods of the year (*Machias & Tsimenidis, 1995*). This study used samples collected at both the beginning (October) and closer to the end (April) of the reproductive period, but apart from the transitory higher gonad volume at hydration (see previously), mean relative gonad volumes are reported to be similar for males and females throughout the spawning season off the Portuguese coast (*Nunes et al., 2011*). For the Barents Sea capelin, although there is a sexual dimorphism in the species life-history (males follow a semelparous batch-spawning strategy and have a lower gonadosomatic index than females which are iteroparous), the effect of gonad weight on the swimbladder size was not significant in both sexes (*Jørgensen, 2003*). In the oyster toadfish, there is a sexual dimorphism in swimbladder size, swimbladders being larger in males, but the authors provide evidence that such differences are related to the function of the swimbladder in sound production (males would produce sounds to attract females for mating) (*Fine, McKnight Jr & Blem, 1995*).

Depth related changes in swimbladder size have already been described for herring (*Blaxter, Denton & Gray, 1979*; *Ona, 1990*; *Fässler et al., 2009*) and are postulated to occur in other physostomous clupeoids too like the Japanese anchovy, *Engraulis japonicus* (*Zhao, Wang & Dai, 2008*). *Balan (1961)* describes for oil-sardine *Sardinella longiceps* on the south-west coast of India that the presence of bottom shoals can often be judged by the emergence of tiny bubbles from the sea-bottom that break-up at the water surface. The author suggests that these bubbles are released from the swimbladder which practically facilitates the fish to sink-down to the sea bed.

The Atlantic sardine is a pelagic fish that performs extended diurnal vertical migrations (*Zwolinski et al., 2007*). As acoustic surveys are often operated on a 24-h basis, the observed depth-dependence of swimbladder size may have direct effect on fish target-strength, TS, and consequently on the resulting biomass estimates. Similar results for depth dependence of TS were provided by *Zhao, Wang & Dai (2008)* for the Yellow Sea anchovy, *E. japonicus*; the authors further suggested that TS values would be elevated by 2 dB when fish migrate from 50 m deep during daylight to 20 m deep at night resulting in a 58% difference between daytime and nighttime biomass estimates. In that respect, our method may provide an easy means for measuring swimbladder size in parallel with body size

measurements in acoustic surveys in order to make the appropriate calibrations in the analysis of backscaterring energy.

## ACKNOWLEDGEMENTS

Drs Yorgos Stratoudakis, and Dr. Alexandra Silva are greatly thanked for providing help at various steps of this work and D. Morais is thanked for the acquisition of sardine biological data during the cruises. The work of S Michou was partly done within the framework of a collaboration between the School of Biology AUTH and IPMA under the Erasmus Placement Programme.

### Funding

The present work was operated within the framework of IPMA's acoustic surveys funded by the Portuguese biological sampling programme integrated in the EU Data Collection Framework (DCF). The funders had no role in study design, data collection and analysis, decision to publish, or preparation of the manuscript.

### Grant Disclosures

The following grant information was disclosed by the authors:
Portuguese biological sampling programme integrated in the EU Data Collection Framework (DCF).

### Competing Interests

The authors declare there are no competing interests.

### Author Contributions

- Kostas Ganias conceived and designed the experiments, performed the experiments, analyzed the data, contributed reagents/materials/analysis tools, wrote the paper, prepared figures and/or tables, reviewed drafts of the paper.
- Stella Michou contributed reagents/materials/analysis tools.
- Cristina Nunes performed the experiments, contributed reagents/materials/analysis tools, reviewed drafts of the paper.

### Animal Ethics

The following information was supplied relating to ethical approvals (i.e., approving body and any reference numbers):

The present study involves routine biometric measurements done in surveys of small pelagic fishes. No relevant permit was required neither in my department (School of Biology, Aristotle University of Thessaloniki) nor in the Institute (IPIMAR, Portugal) that collected this data.

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
