# Peer review of "A field based study of swimbladder adjustment in a physostomous teleost fish"

_PeerJ, doi:10.7717/peerj.892_

## Round 0.1 · original submission · Major Revisions

As per the reviewers, the manuscript has to be improved and the section of Methods have to be completed according to the reviewer's suggestions.

·

Basic reporting

ok

Experimental design

Missing information. See general comments.

Validity of the findings

I am not convinced that the measurements reflect in situ values of swimbladder dimensions.

Additional comments

Although it is generally well written and required a lot of work, I am uncomfortable with Ganias et al’s paper for a number of reasons. First and foremost, it is difficult to believe that measurements in air accurately reflect in situ values since gas volumes, hence ventral cross-sectional areas, are likely to be affected by hydrostatic pressure and ascent, particularly since some bladders were deflated and others distended. I also don’t understand the constrictions in certain bladders and how they affect measurements. The authors do not seem to be aware of papers indicating that herring can make FRT sounds (fast repeated ticking) by eructating bubbles from their anus, reflecting gas loss from the bladder of fish in shallow water (Wahlberg and Westerberg, 2003; Wilson et al. 2004).
There is information that should likely have been included in the methods. For fish processing: were the fish processed immediately on deck, fixed or frozen? Time of trawling was not mentioned, and there could be day-night effects since fish are deeper in the day. Fish sex was not mentioned as a factor although ovary size was prominently considered. The size ranges in the different size categories was never delineated. Taking pictures at different angles does not say the number of pictures or the specific angles.
The authors use some fancy statistics, which makes the results complex and difficult to follow. I may be doing them an injustice, but I would rather see a number of simple linear regressions using fish measurements rather than categories. Measurements of bladder length could be plotted against total length or bladder weight against fish weight separately for males and females, depth etc. This should allow linear regressions that could be compared simply with ANCOVA. Multiple regressions could be used in some cases such as weight and gonad weight in females. Such measurements would bypass problems with pressure affects and allow the authors to tell a simple story if there is one. Fig. 5 indicates a large variation within the sample, and the individual comparisons I suggest will hopefully reduce much of this variation and add to our knowledge of what happens to these fish with sex, gonad development, depth, etc.
Minor points
L. 23 Swimbladder gas affects the acousticolateralis system in clupeids and those with specific connections between the bladder and the ears (Yan et al. 2000), but it will not be important in most teleosts.
L. 34. The authors make it sound as if there is something wrong with measuring swimbladder volume directly. Just say that determining bladder shape requires morphological measurements.
L. 45. Opening is to the alimentary canal and then to the anus.
L. 47. Subtle should probably be subject to…, and diurnal, meaning day-active is misused. The correct word is diel.
L. 61. Methods for maturity, fat index and stomach fullness need to be detailed. Some of this is mentioned later. Please put it in one place.
L. 72, Specify angles.
L. 77. Delete “a total of” and “finally.”
L. 79. Delete the… in each of the aforementioned specimens… in more detail.. that were collected during the surveys. All of this is obvious.
L. 83. Delete Apart from analyzing its cross sectional area variability in.
L. 88. Delete as the value approaches 0.0 it indicates and replace with lower values indicate.
L. 98. Delete among the aforemented descriptors.
L. 189. Blaxter’s statement that gas loss is impossible without activating the voiding reflex does not mean that it was not activated.
Michael L. Fine
Reference List

Wahlberg MH, Westerberg H (2003) Sounds produced by herring (Clupea harengus) bubble release. Aquat.Living Resourc., 16, 271-275.
Wilson B, Batty RS, Dill LM (2004) Pacific and Atlantic herring produce burst sounds. Proc.Roy.Soc.Lond.B (Suppl), 271, 95-97.
Yan HY, Fine ML, Horn NS, Colon WE (2000) Variability in the role of the gasbladder in fish audition. J.Comp.Physiol.A, 187, 371-379.

Reviewer 2 ·

Basic reporting

no comments

Experimental design

no comments on almost all areas except that in the manuscript it was not clear if both males and females were used in the experiment and how sex ratio was distributed among samples. To provide a comparison between the two sex in the swim bladder volume would be useful also to further supports the main findings.

Validity of the findings

Data and statistical treatment is rigorous and the validity off results enough, however some aspect related to the gonadic maturity patterns in males and females should be still addressed.
The authors tested several important variables for haul depth, but to exclude other possibile causes of the relationship between swimbladder and depth, reproductive parameters like sex ratio and maturity stage patterns have to be taken into account. More specifically in the manuscript (at least in the discussion section) it would be discussed the possibility that a sexual/maturing segregating behaviour of spawning females may affects the results.

Additional comments

The manuscript address in a new way an important issue: the swim bladder volume variability of an important species whose stock are evaluated mainly by acoustic methods. The work presented could be replicated in other species and other areas in order to increase the acoustic biomass estimation.

---

## Round 0.2 · accepted · Accept

We hope you consider PeerJ for your next works.